# A Study on Wetland Cover Map Formulation and Evaluation Using Unmanned Aerial Vehicle High-Resolution Images

**Tai Yang Lim, Jiyun Kim, Wheemoon Kim and Wonkyong Song ***

Green & Landscape Architecture, Dankook University, Cheonan 31116, Republic of Korea;
hyperspectral0811@gmail.com (T.Y.L.); animal414@naver.com (J.K.); wheemoon_@naver.com (W.K.)
* Correspondence: wksong@dankook.ac.kr; Tel.: +82-41-550-6273

**Abstract:** Wetlands possess significant ecological value and play a crucial role in the environment. Recent advancements in remote exploration technology have enabled a quantitative analysis of wetlands through surveys on the type of cover present. However, the classification of complex cover types as land cover types in wetlands remains challenging, leading to ongoing studies aimed at addressing this issue. With the advent of high-resolution sensors in unmanned aerial vehicles (UAVs), researchers can now obtain detailed data and utilize them for their investigations. In this paper, we sought to establish an effective method for classifying centimeter-scale images using multispectral and hyperspectral techniques. Since there are numerous classes of land cover types, it is important to build and extract effective training data for each type. In addition, computer vision-based methods, especially those that combine deep learning and machine learning, are attracting considerable attention as high-accuracy methods. Collecting training data before classifying by cover type is an important factor that which requires effective data sampling. To obtain accurate detection results, a few data sampling techniques must be tested. In this study, we employed two data sampling methods (endmember and pixel sampling) to acquire data, after which their accuracy and detection outcomes were compared through classification using spectral angle mapper (SAM), support vector machine (SVM), and artificial neural network (ANN) approaches. Our findings confirmed the effectiveness of the pixel-based sampling method, demonstrating a notable difference of 38.62% compared to the endmember sampling method. Moreover, among the classification methods employed, the SAM technique exhibited the highest effectiveness, with approximately 10% disparity observed in multispectral data and 7.15% in hyperspectral data compared to the other models. Our findings provide insights into the accuracy and classification outcomes of different models based on the sampling method employed in spectral imagery.

**Keywords:** hyperspectral; spectral angle mapper; support vector machine; neural net; drone; wetland classification

## 1. Introduction

Wetlands are essential components of ecosystems and hold significant ecological value, including ecological biodiversity, water quality restoration, habitat provision, and climate regulation [1]. Given the global environmental changes caused by climate change, wetland conservation has gained increasing importance [2,3]. Therefore, classifying the habitat and environmental conditions of species within wetland ecosystems is 2, crucial [4]. To achieve this, effective methods and approaches for data collection and analysis pertaining to wetland coverage are needed.

Wetland surveys play a vital role in comprehending and conserving complex ecosystems, thus contributing to the sustainable management of wetlands [5]. While traditional surveys rely heavily on human involvement, recent advances in remote sensing (RS) have facilitated surveys and data collection in a more cost-effective and efficient manner [6,7]. To

Remote-based surveys allow for non-invasive approaches and quantitative data acquisition [8,9]. Particularly, unmanned aerial vehicles (UAVs) have emerged as effective tools for acquiring various types of remote sensing data [10]. High-resolution imagery captured by UAVs encompasses a wide electromagnetic spectrum, including visible and infrared light [11]. Spectral-based data acquisition has been employed in research focused on classification and detection due to its ability to acquire biochemical and biophysical parameters [12]. Integrating spectral data with RS-based studies enables the mapping of large-scale target sites [13]. Mapping research utilizing high-resolution spectral data, such as multispectral and hyperspectral data, proves valuable in monitoring wetlands composed of diverse surface textures [14]. Furthermore, the use of UAV imagery has expanded to various fields, including ecosystem monitoring, environmental assessment, land use analysis, and mapping [15,16].

The utilization of high-resolution UAV imagery is an effective approach for land cover classification; however, several factors must be considered in the classification process [17]. Given that these approaches rely on captured images, factors such as spatial resolution, altitude, shooting time, classification techniques, and data sampling must be accounted for during drone flights and image acquisition [18]. Among these factors, data sampling and algorithm selection play a crucial role in the classification methods [19,20]. Currently, statistical analysis and machine learning methods such as support vector machines (SVMs), spectral angle mappers (SAMs), and k-nearest neighbors (KNNs) have been proven to be effective [21–23]. Nevertheless, traditional learning methods often struggle to meet high classification accuracy, posing limitations for analysts [24]. Therefore, recent research has embraced methods that leverage computer vision, such as machine learning and deep learning, which require extensive training data [25]. This approach offers the advantage of achieving high-accuracy results by processing complex and functional data compared to conventional learning methods [26]. Therefore, there are ongoing efforts to enhance the efficiency and accuracy of classification through computer vision techniques, including machine learning and deep learning [27].

A substantial amount of data is needed for effective classification when utilizing computer vision for land cover classification. However, obtaining quantitative data can be challenging due to the wide range of variations within each class [28]. Therefore, quantitative data extraction plays a critical role as a fundamental analysis component. This study proposes an effective classification method utilizing image cluster techniques and pixel purity index (PPI)-based end member extraction to evaluate quantitative data composition.

Various classification methods exist for analyzing images based on the classification criteria by establishing data sampling and reference libraries [29,30]. Among these methods, the image cluster technique involves dividing the target site into several clusters and assigning a cluster to each class for data sampling [31,32]. Additionally, the endmember technique aims to extract the purest value within an image. The PPI technique calculates the n-D scatterplot by repeatedly projecting it onto any unit vector, defining the pixel with the most repeated projection as the endmember [33,34]. Since the spectra generated through these two sampling techniques represent data values for significant components in the image, a comparison between them is necessary, as they have the potential to influence the classification outcomes. Furthermore, to evaluate the accuracy based on the training data configuration suitable for each class, the accuracy was assessed through a supervised classification by comparing classes.

In this study, data were acquired using unmanned aerial vehicles equipped with multispectral and hyperspectral camera sensors, thus facilitating wetland mapping. Moreover, the data sampling method was evaluated to ensure effective wetland mapping, along with an assessment of the classification method aligned with the chosen technique.

## 2. Materials and Methods

### 2.1. Study Site

This study was conducted at Ansan Reed Wetland, located in Ansan-si, Korea. The wetland is situated at 37°16′21.32 N, 126°50′21.64 E, with a total area of 1,037,500 m². Since its establishment in September 1997, the wetland has been designated as a protected area and managed accordingly. Ansan Reed Wetland serves as a wetland park, aimed at treating non-point pollutants and providing habitats for organisms in the surrounding basin. Currently, the wetland supports the distribution of 290 species of vegetation and accommodates 150 species of migratory birds [35].

For the purpose of this study, a specific area within Ansan Reed Wetland was selected and photographed, thus capturing the present status of the ecosystem. The filming was conducted in an area measuring 11,059 m², which was selected to represent the complex classes of the entire target site's ecosystem.

### 2.2. UAV Flight and Data Acquisition

On 24 October 2022, a UAV survey was conducted over Ansan Reed Wetland (Figure 1). The target site aerial imagery was acquired using DJI Matrice 300 RTK and Matrice 600 drones (http://www.dji.com (accessed on 2 October 2022), see Table 1). For filming, DJI Zenmuse L1 and MicaSense Redge-MX cameras were attached to the Matrice 300 RTK drone, ensuring a coordinate error of less than 1 cm through the DJI D-RTK2 (Real-Time Kinematic) method for orthographic images, multispectral, and data collection [36,37]. The Matrice 600 drone was equipped with a Headwall Nano-Hyperspec VNIR camera to obtain hyperspectral data [38]. The hyperspectral VNIR camera acquired DEM data through the Velodyne LiDAR option (https://www.headwallphotonics.com/products/vnir-400-1000nm (accessed on 2 October 2022)). During the acquisition of hyperspectral data, Trimble R4s was utilized for post-processed kinematic (PPK) GNSS correction, thus ensuring that the geometric accuracy aligned with that of D-RTK2.

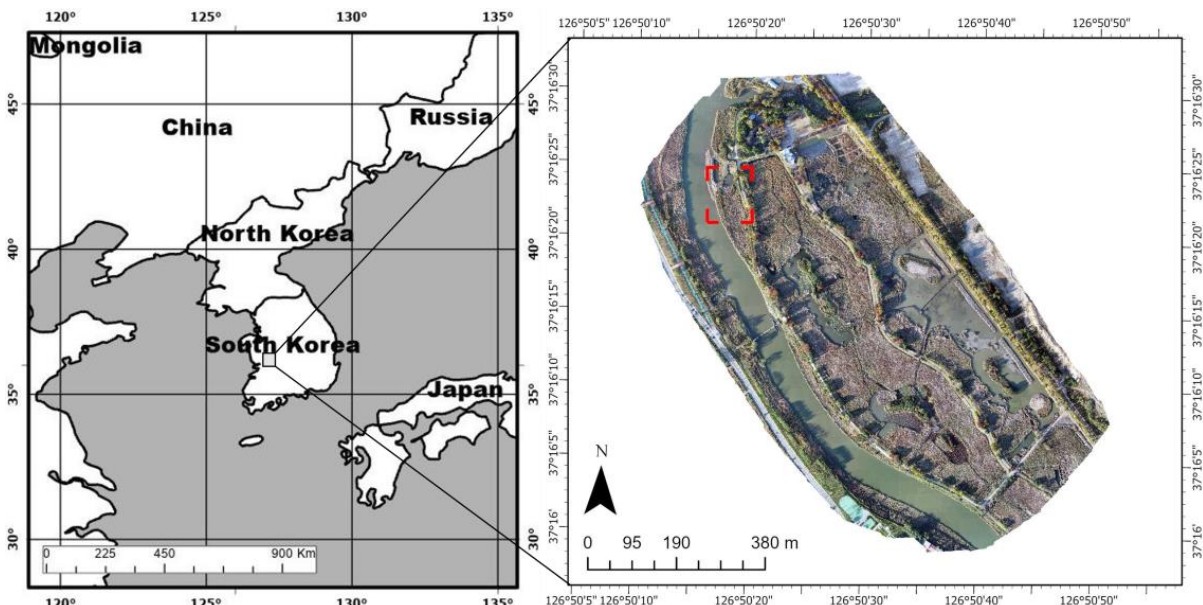

**Figure 1.** Ansan Reed Wetland, located in Ansan, South Korea. The red area represents the location of the Ansan Reed Wetland. The image was captured using a DJI Zenmuse L1 sensor mounted on a Matrice 300 RTK drone.

Filming was conducted at a 100 m altitude, primarily at noon to minimize the impact of shadows. The timing was carefully selected, considering the solar altitude angle, and to ensure effective data acquisition, an 80% longitudinal/transverse overlap was maintained. Filming took place on clear days with minimal wind interference to optimize the results.

**Table 1.** Specifications of the DJI Zenmuse L1, MicaSense Rededge-MX, and Headwall Nano-Hyperspec VNIR sensors used for UAV imaging.

| Instrument | L1 | Redege-MX | Nano-Hyperspec VNIR |
|---|---|---|---|
| Bands | 3 | 5 | 270 |
| FOV | 95° | 47.2 ° | 50.68° |
| Wavelength range (nm) | - | 475, 560, 668, 717, 840 nm | 400–1000 nm |
| GSD (100 m) | 2.73 cm/pixel | 9.26 cm/pixel | 11 cm/pixel |
| Supported aircraft | Matrice 300 RTK | Matrice 300 RTK | Matrice 600 |

### 2.3. UAV Image Processing

The data acquired by the UAVs underwent various processing steps. The orthographic image was generated by mosaicking the captured images using DJI Terra software (DJI, Shenzhen, China). For the correction of multispectral data, real-time light measurements were conducted using the DLS 2 sensor, and the values obtained from the Calibrated Reflection Panel (CRP) were used to account for solar angle and lighting conditions during image acquisition. Mosaicking and radiometric correction were performed using the Pix4D Mapper software (Pix4D SA, Lausanne, Switzerland) [39,40]. Afterward, each extracted band was aligned with the corresponding multispectral image using the band stack function [41].

Hyperspectral images were acquired using the Hyperspec III sensor from Headwall (Headwall Photonics, Inc., Bolton, MA, USA) [42]. During image processing, a correction factor was derived by utilizing a correction tarpaulin, which was pre-treated to have a reflectance of 56%. This correction factor was applied for radiometric correction [43,44]. Geometric correction was performed based on a digital elevation model (DEM) and GPS/IMU data generated through LiDAR technology integrated with the Nano-Hyperspec VNIR sensor. To reduce noise present in the spectrum, smoothing was applied using the Savitzky–Golay filter [45,46].

### 2.4. Field Cover Type Investigation

Field surveys were conducted in the study target site to investigate the main land class types. During the field survey, the field classes were recorded by directly examining the site and referring to the Aerial image of DJI L1. The survey took place on 17 October 2022. Ten classes were identified based on the on-site investigation, including conifers (pines), two types of broadleaf trees (red maple and cherry trees), mud, grass (other vegetation), sand, road (concrete), water bodies, reeds, and pergola (urethane). The land cover types determined from the field survey were digitized using the ArcGIS Pro software (Environmental Systems Research Institute, Inc., Redlands, CA, USA). These digitized data served as reference and verification data for evaluating the accuracy of the classification models.

### 2.5. Classification Algorithm

The classification of classes within the study site was performed using widely used classification models, including SAM (Spectral Angle Mapper), SVM (Support Vector Machine), and ANN (Artificial Neural Network). SAM identifies and classifies similarities by analyzing the interior of the vector in the n-dimensional space between the selected pixel or reference spectrum and the spectrum of the image [47–49]. One advantage of SAM is its ability to achieve effective classification by reducing the dimensionality of the data and classifying the image based on the direction of the angle, regardless of vector size [50]. In SAM classification, a smaller angle indicates a higher degree of agreement with the

reference spectrum, whereas angles greater than a set threshold are not classified. The SAM classification process is described in Equation (1) below:

$$\alpha = \cos^{-1}\left[\frac{\sum_{i=1}^{nb} t_i r_i}{\left(\sum_{i=1}^{nb} t_i r_i\right)^{\frac{1}{2}}\left(\sum_{i=1}^{nb} t_i r_i\right)^{\frac{1}{2}}}\right] \tag{1}$$

where $t$ represents the spectrum of the pixel, $r$ represents the reference spectrum pixel, $\alpha$ represents the spectral angle between $t$ and $r$, and $n$ represents the number of bands. In this study, classification was performed using a threshold value of 0.4 [51].

SVM is a statistics-based supervised classification learning method that aims to maximize margins by constructing reference training data in the form of margins on a hyperplane [52–54]. However, given that the linear separation of data has its limits, SVM maps and separates the data in a high-dimensional feature space using various kernel methods [55]. Furthermore, SVM allows for the classification of multiple classes by conducting pairwise classification, and the adjustment of parameters can help reduce misclassifications [56]. In this study, SVM utilized a radial basis function (RBF) kernel for pairwise classification [57] (2).

$$kxixjK = \exp\left(-g\left|\left|x_i - x_j\right|\right|^2\right),\ g > 0 \tag{2}$$

Furthermore, the classification method employed various input parameters, including the values of the gamma ($\gamma$) kernel functions, pyramid levels, penalty parameters, and classification probability thresholds. For the SVM classification, the penalty parameter was set to its maximum value (100) to minimize misclassifications. The gamma ($\gamma$) kernel function value was set to 0.007, and the classification probability threshold was set to 0 to ensure proper classification of the training data into each class [58].

Artificial neural network (ANN) is a multi-layer neural network classification technique comprising multiple layers. In this study, the ANN configuration utilized a feedforward-based backpropagation algorithm with supervised learning, analyzing the data using a chain structure [59,60]. Specifically, a segment-based U-Net algorithm was employed to establish corresponding areas of interest (ROIs) and convert them into training data. Deep learning requires the configuration of detailed parameters, including training contributions, training rates, training end values, training iterations, and hidden layer configurations [61]. In this paper, the parameters with the highest accuracy and reliability for neural networks were chosen as a reference, considering previous studies. Here, the training contribution was set to 90%, the training rate was set to 90%, the training exercise was set to 0.1, the RMSEC (root mean square error of calibration) was set to 0.08, the hidden layer was set to 1, and the training repetitions were set to 1000 times.

### 2.6. Training Data Processing

### 2.6.1. Pixel Sampling

The process of generating training data involves four stages. Firstly, since each image has a different spatial resolution, resampling was performed to ensure uniform pixel size and coordinates for both multispectral and hyperspectral data. Secondly, object image segmentation was conducted to divide the clusters within the images. Thirdly, the unsupervised classification algorithm ISO-DATA was applied to organize the training data based on the average values within the appropriate class-defined segments. ISODATA performs classification using pixel thresholds across the entire image and conducts class-specific classification based on standard deviation, distance threshold, and other factors [62,63]. This approach facilitated pixel-based classification within the target site. The segmented images were saved as shapefiles, class-specific configurations were edited using ArcGIS Pro, and normalization was performed through masking using the ENVI 5.6 software. Finally, pixel data for each class were extracted from the original image based on the corre-

sponding classified image and unsupervised classification results (Table 2). The collected pixel sampling data were then utilized as training data for subsequent classification models. The preprocessing workflow incorporating pixel sampling is illustrated in Figure 2. The ENVI 5.6 (Exelis Visual Information Solutions, Inc., Boulder, CO, USA) and ArcGIS Pro software were employed for data extraction and processing.

**Table 2.** Number of training and validation data samples for each class.

| Class | Coniferous Trees | Broadleaf Trees | Mud | Grass | Sand | Road | Water | Reed | Maple | Pergola |
|---|---|---|---|---|---|---|---|---|---|---|
| Training Data | 1000 | 1000 | 1000 | 1000 | 1000 | 1000 | 1000 | 1000 | 1000 | 1000 |
| Validation data | 200 | 200 | 200 | 200 | 200 | 200 | 200 | 200 | 200 | 200 |

Unit: Pixels.

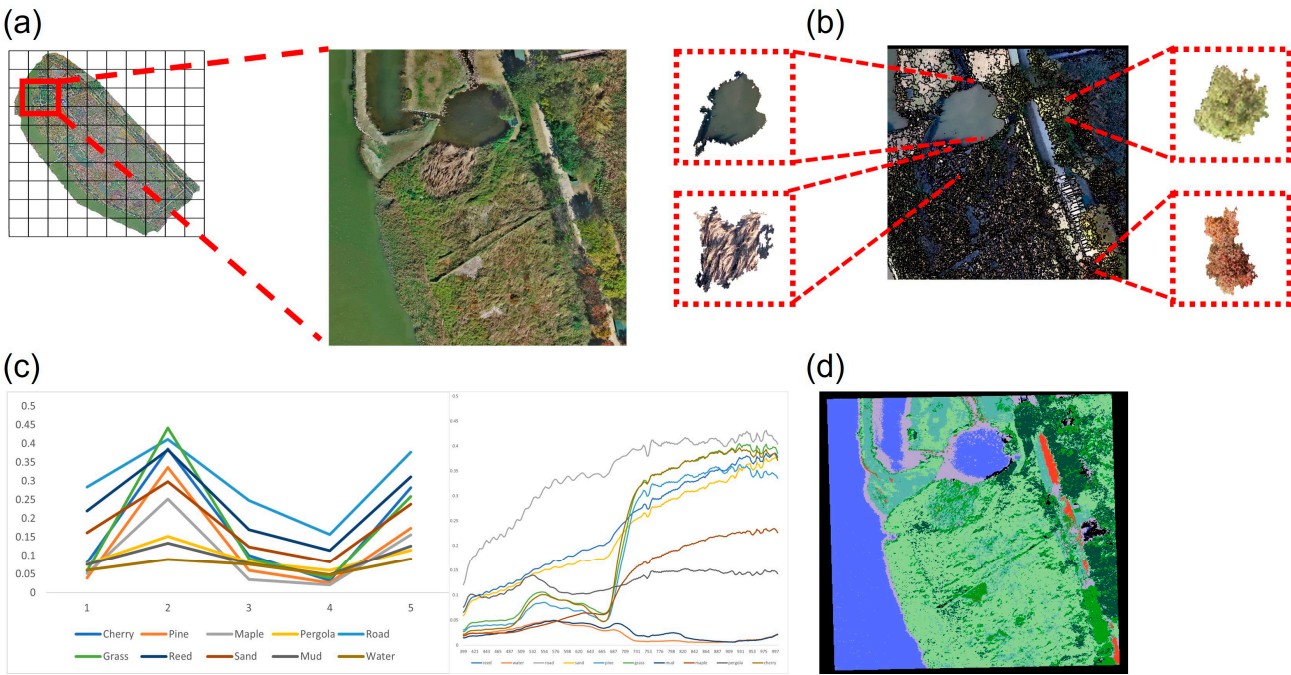

**Figure 2.** Pixel sampling workflows. (**a**) Identification of target areas within the image; (**b**) collection of pixels belonging to each class within the target areas; (**c**) extraction of average spectra from the collected pixels and setting of ROIs; (**d**) extraction of results based on the defined areas and spectral information.

### 2.6.2. Spectral Sampling

Endmember extraction using the PPI was performed in three main stages. Firstly, data dimensional reduction was carried out using a minimal noise fraction (MNF) transformation [64]. MNF transformation helps to identify pure pixels with minimal noise and minimizes distortion caused by noise in the pixels [65,66]. Through this process, the MNF transformation determined the location of pure pixels, after which the PPI technique was applied to process the data through 10,000 repetitions [67]. The extracted endmembers were validated using the PPI technique. Six bands were selected from the five bands available in the multispectral image, whereas forty-eight bands were selected from the two hundred and seventy-three bands in the hyperspectral image. This endmember quantification approach was adopted because 10 classes were identified through the field survey and the number of multispectral images was limited, whereas there were many numerous hyperspectral images. Pearson correlation analysis was conducted by applying it to each image to select the band that had the highest correlation with the selected class among the endmembers extracted using the PPI technique (Figure 3).

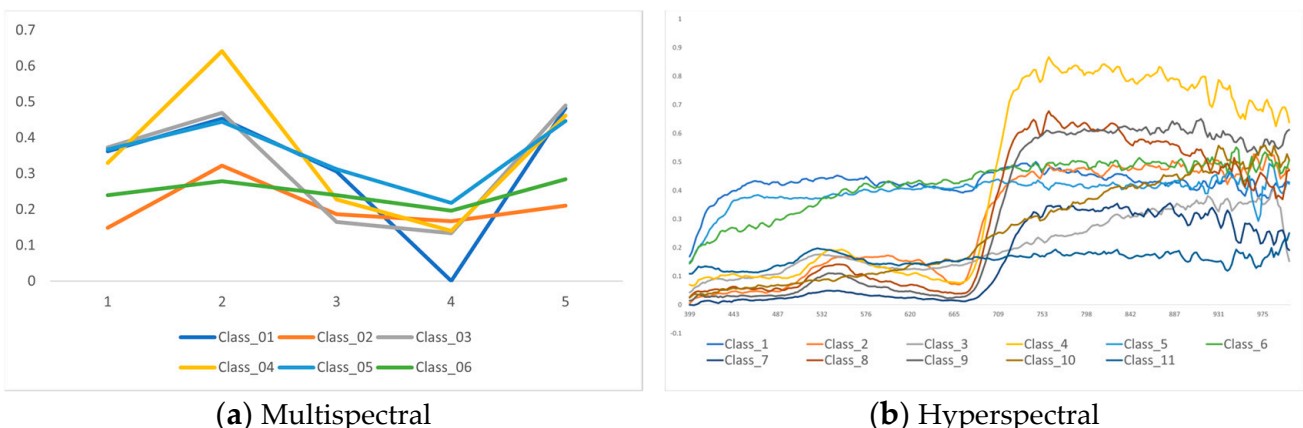

(**a**) Multispectral      (**b**) Hyperspectral

**Figure 3.** Endmember extraction using the PPI technique ((**a**) multispectral, (**b**) hyperspectral).

Class classification between the images obtained through the selected endmember and pixel sampling methods was performed to validate the effectiveness of the data sampling approaches. Furthermore, SAM classification was conducted to compare the pixel sampling data and examine the correlation and classification accuracy between the two datasets. A comparative analysis was then conducted using the verification data.

### 2.7. Accuracy Assessment

2.7.1. Producing Verification Data

To construct accurate verification data for image validation, an on-site survey was conducted in five test beds on 17 and 24 October 2022. The verification data were generated based on the object segmentation image. High-resolution orthophotos and field verification were used to ensure the correct classification of each pixel-based class. A random 10 m$^2$ grid was generated within each of the five squares to establish the verification data. The data were edited using ArcGIS Pro, and segment classification was performed using the orthophotos. The class was then edited using the corresponding image (Figure 4).

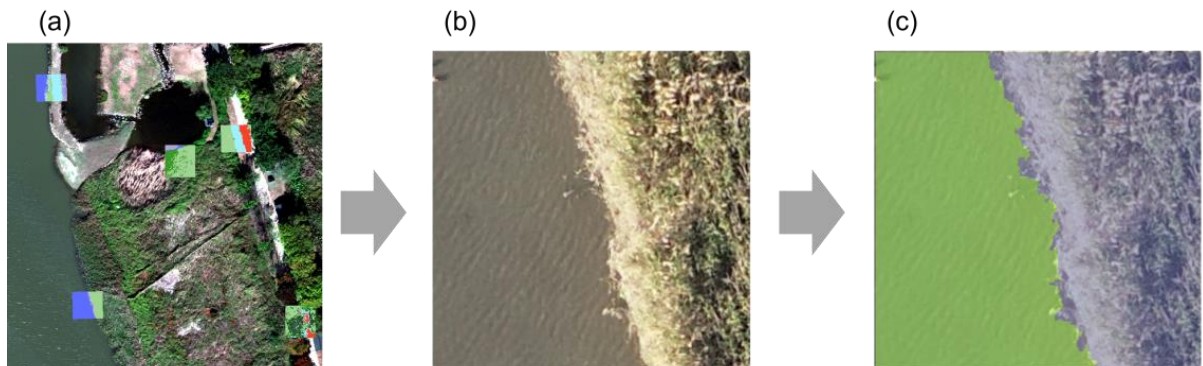

**Figure 4.** Workflow of verification data generation (**a**) overview image showing the five testbeds areas designated for verification within the target area, (**b**) detailed image of the testbeds, (**c**) image of the verification data construction process.

2.7.2. Evaluating Validation Data Accuracy

To assess the accuracy of each land cover classification method using UAV images, data verification was conducted using a confusion matrix based on Ground Truth, Kappa coefficients, and F1 scores [68]. The Kappa coefficient is used as a traditional method to measure reliability, but it is not appropriate for an unbalanced distribution of classes, so we further evaluated reliability using F1 scores [69,70]. The F1 score is evaluated using the confusion-matrix-based precision and recall values [71]. The evaluation involved dividing

the entire image into $10 \times 10$ square meters. A land cover map was generated using a classifier for five random points, after which the accuracy was verified by comparing the resulting map with the reference image. The comparison between the two images was evaluated based on overall accuracy (OA) and reliability using the Kappa coefficient and F1 score. This allowed for an assessment of the overall accuracy for each image and classification method [72].

$$\text{Precision} = \frac{TP}{TP + FP} \text{Recall} = \frac{TP}{TP + FN} \text{F1 Score} = \frac{2 * Presicion * Recall}{Precision + Recall} \quad (3)$$

## 3. Results

### 3.1. Classification Technique Method Results

The training data were constructed using UAV-acquired images, and the classification was performed using SAM, SVM, and ANN (Figure 5). The resulting classification images were compared with the verification data to evaluate the accuracy and reliability of each classification method (Tables 3 and 4). The hyperspectral and multispectral classification results showed that SAM achieved an accuracy of 91.906% to 80.364%, SVM achieved 84.788% to 77.006%, and ANN achieved 84.712% to 70.750%. In hyperspectral and multispectral images, the SAM method achieved the highest accuracy compared to other methods. The SAM showed excellent performance in spectral image classification, and the Kappa coefficient and F1 score were checked to compare the reliability of the verification. In the hyperspectral case, the Kappa coefficient was 0.88 for SAM, 0.76 for SVM, 0.77 for ANN, and the F1 score was 0.836 for SAM, 0.764 for SVM, and 0.769 for ANN. In the multispectral case, the Kappa coefficient was 0.720 for SAM, 0.666 for SVM, 0.612 for ANN, and the F1 score was 0.719 for SAM, 0.606 for SVM, and 0.675 for ANN. We show that SAM achieves higher verification accuracy compared to SVM and ANN in both the multispectral and hyperspectral data. In addition, when comparing reliability through the Kappa coefficient and F1 score, all SAM techniques confirmed a confidence of 0.7 or more. Overall, the assessment results revealed that hyperspectral images provided higher verification accuracy than hyperspectral images, and the SAM analysis technique was more effective for classification purposes.

**Table 3.** Results of verifying classification methods (SAM, SVM, and ANN) using multispectral images.

| Multispectral | SAM | | | SVM | | | ANN | | |
|---|---|---|---|---|---|---|---|---|---|
| | OA (%) | Kappa | F1 Score | OA (%) | Kappa | F1 Score | OA (%) | Kappa | F1 Score |
| 1 | 92.96 | 0.87 | 0.859 | 95.38 | 0.89 | 0.678 | 96.62 | 0.93 | 0.773 |
| 2 | 93.43 | 0.92 | 0.941 | 80.92 | 0.75 | 0.778 | 70.96 | 0.62 | 0.744 |
| 3 | 91.44 | 0.85 | 0.817 | 89.25 | 0.83 | 0.695 | 63.27 | 0.51 | 0.725 |
| 4 | 64.04 | 0.48 | 0.364 | 74.51 | 0.55 | 0.407 | 75.16 | 0.65 | 0.606 |
| 5 | 59.95 | 0.48 | 0.613 | 44.97 | 0.31 | 0.470 | 47.74 | 0.35 | 0.527 |
| Total (average) | 80.364 | 0.720 | 0.719 | 77.006 | 0.666 | 0.606 | 70.750 | 0.612 | 0.675 |

**Table 4.** Results of verifying classification methods (SAM, SVM, and ANN) using hyperspectral images.

| Hyperspectral | SAM | | | SVM | | | ANN | | |
|---|---|---|---|---|---|---|---|---|---|
| | OA (%) | Kappa | F1 Score | OA (%) | Kappa | F1 Score | OA (%) | Kappa | F1 Score |
| 1 | 98.28 | 0.95 | 0.910 | 97.90 | 0.95 | 0.911 | 96.95 | 0.93 | 0.866 |
| 2 | 99.83 | 0.99 | 0.996 | 99.40 | 0.99 | 0.993 | 98.97 | 0.98 | 0.988 |
| 3 | 83.22 | 0.74 | 0.455 | 86.43 | 0.79 | 0.846 | 92.00 | 0.89 | 0.655 |
| 4 | 99.36 | 0.99 | 1.000 | 91.31 | 0.85 | 0.917 | 93.45 | 0.87 | 0.904 |
| 5 | 78.84 | 0.70 | 0.819 | 48.90 | 0.24 | 0.533 | 42.19 | 0.16 | 0.433 |
| Total (average) | 91.906 | 0.874 | 0.836 | 84.788 | 0.764 | 0.840 | 84.712 | 0.766 | 0.769 |

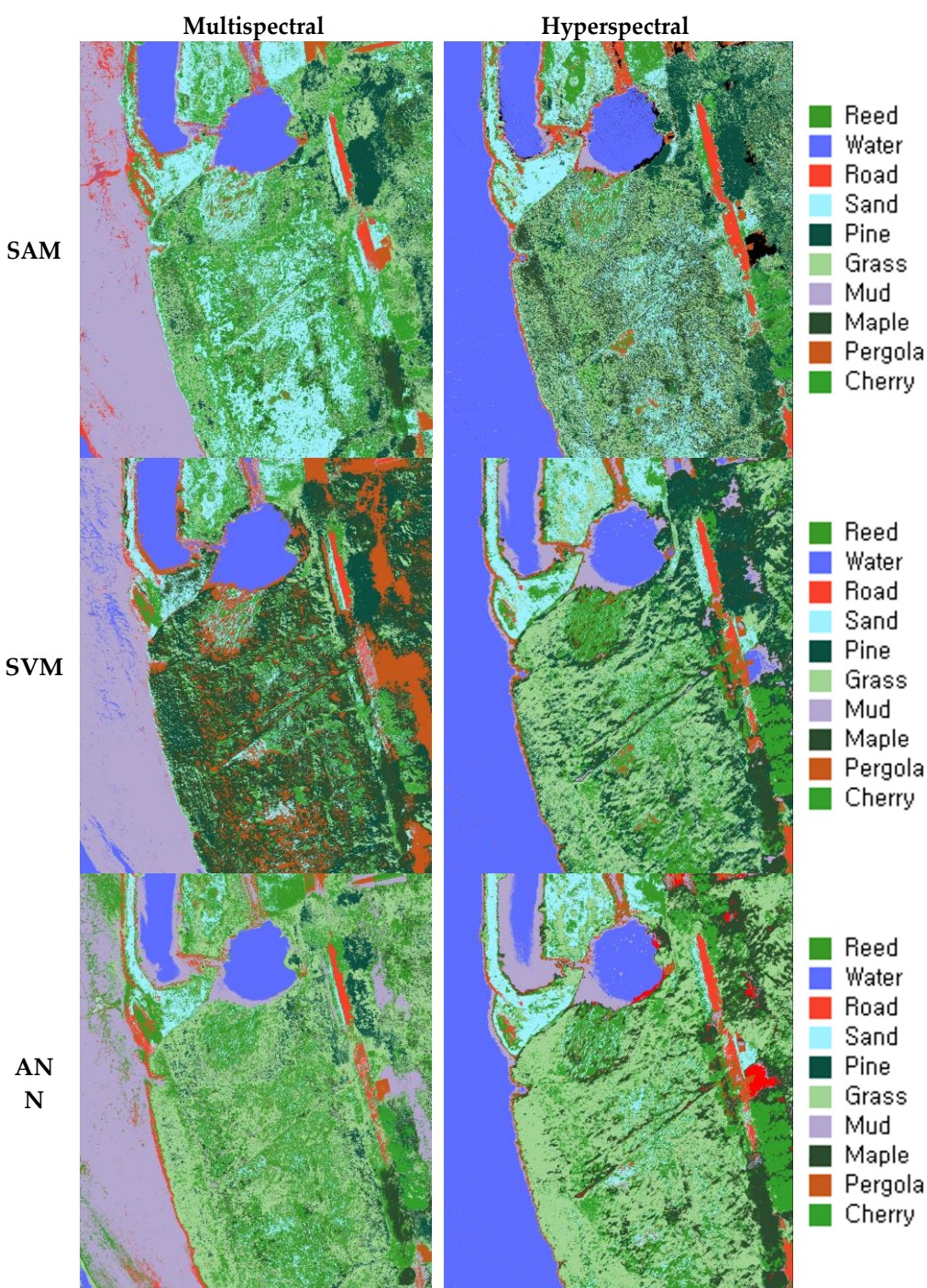

**Figure 5.** SAM, SVM, and ANN classification via UAV images (multispectral, hyperspectral).

As a result of calculating the area according to each classification method, it was confirmed that the average grass (19.42%) and water (15.93%) areas of the hyperspectral and multispectral images were the largest, followed by maple (12.27%), mud (11.64%), reed (9.78%), pine (7.94%), cherry (7.37%), and roads (12%) (Table 5). Unclassification refers to groups that were not classified at the time of the categorization, and the in-image values have been identified as ANN (0.86%) and SVM (1.23%) during hyperspectral division. In the hyperspectral images, the largest class differences in ANN, SVM, and SAM were identified as reeds (±6.58%), maple (±6.48%), and sand (±6.14%), and ANN was characterized by differences in vegetation-related groupings such as pine (±13.96%) and grass (±7.48%). In the case of multispectral images, there were many variances in class proportions when classifying ANN, SVM, and SAM, and most notably, ANN had the largest difference in

reeds (±22.62%), grass (±23.82%), maple (±21.31%), SVM sand (±20.26%), and cherry (±12.32%) compared to other classes.

**Table 5.** Results of verifying the area for the classification methods (SAM, SVM, and ANN) using hyperspectral and multispectral images.

| | Hyperspectral | | | Multispectral | | |
|---|---|---|---|---|---|---|
| **Class** | **ANN** | **SVM** | **SAM** | **ANN** | **SVM** | **SAM** |
| Unclassified | 0.858% | 1.229% | 0.000% | 0.002% | 0.002% | 0.000% |
| Reed | 11.083% | 6.056% | 12.635% | 24.385% | 2.741% | 1.769% |
| Water | 24.421% | 25.889% | 25.889% | 4.856% | 6.508% | 8.012% |
| Road | 1.417% | 3.324% | 1.251% | 2.644% | 2.832% | 1.255% |
| Sand | 7.582% | 13.642% | 7.502% | 4.280% | 18.815% | 6.892% |
| Pine | 0.000% | 11.159% | 13.962% | 4.318% | 7.857% | 10.343% |
| Grass | 27.197% | 18.561% | 19.719% | 30.271% | 14.345% | 6.450% |
| Mud | 4.266% | 1.104% | 4.113% | 21.485% | 19.425% | 19.424% |
| Maple | 13.200% | 15.091% | 8.609% | 4.780% | 5.830% | 26.093% |
| Pergola | 2.038% | 1.193% | 2.805% | 0.581% | 1.689% | 12.125% |
| Cherry | 7.938% | 2.753% | 3.513% | 2.399% | 19.955% | 7.637% |

*3.2. Image Spectrum Comparison*

Using the PPI technique, the difference between the endmember-extracted data and the pixel-sampled data was examined for both multispectral and hyperspectral images (Tables 6 and 7). The Pearson correlation coefficient was calculated, and several associations were observed at a 95% significance level. Among the hyperspectral images, Class_01 exhibited the strongest association with water and mud. Class_02 was associated with cherry, Class_06, and pine. Class_08 was associated with reed, sand, and maple. For the multispectral images, Class_02 showed an association with pine. Class_04 was associated with cherry, maple, pergola, reed, and sand, whereas Class_06 was associated with roads, and Class_08 was associated with mud and water. These results indicate that roads can be classified more accurately based on their distinct spectral characteristics. The relevant classes between end members and pixel sampling are reeds, roads, sand, pine, grass, maple, pergola, and cherries, and water and mud are far apart. Therefore, since pixel sampling is more effective than endmember extraction in identifying components in multiple layers, we show that using that methodology in mapping is efficient for classification.

**Table 6.** Correlation analysis results between endmember and pixel sampling data using hyperspectral images.

| | Class 1 | Class 2 | Class 3 | Class 4 | Class 5 | Class 6 | Class 7 | Class 8 | Class 9 | Class 10 | Class 11 |
|---|---|---|---|---|---|---|---|---|---|---|---|
| Reed | 0.339 | 0.95 | 0.942 | 0.899 | 0.614 | 0.878 | 0.886 | 0.854 | 0.925 | 0.989 | 0.56 |
| Water | −0.101 | −0.808 | −0.745 | −0.882 | −0.228 | −0.505 | −0.877 | −0.862 | −0.884 | −0.815 | −0.262 |
| Road | 0.551 | 0.895 | 0.823 | 0.809 | 0.778 | 0.987 | 0.789 | 0.791 | 0.812 | 0.862 | 0.633 |
| Sand | 0.351 | 0.951 | 0.943 | 0.896 | 0.615 | 0.876 | 0.881 | 0.85 | 0.923 | 0.988 | 0.574 |
| Pine | 0.358 | 0.982 | 0.927 | 0.983 | 0.505 | 0.792 | 0.973 | 0.955 | 0.994 | 0.959 | 0.555 |
| Grass | 0.348 | 0.98 | 0.937 | 0.974 | 0.511 | 0.796 | 0.963 | 0.942 | 0.99 | 0.968 | 0.56 |
| Mud | 0.131 | −0.603 | −0.674 | −0.7 | −0.006 | −0.224 | −0.721 | −0.653 | −0.732 | −0.695 | −0.08 |
| Maple | 0.299 | 0.967 | 0.938 | 0.937 | 0.542 | 0.831 | 0.924 | 0.897 | 0.958 | 0.99 | 0.526 |
| Pergola | 0.451 | 0.893 | 0.858 | 0.916 | 0.473 | 0.725 | 0.916 | 0.908 | 0.919 | 0.829 | 0.744 |
| Cherry | 0.368 | 0.988 | 0.932 | 0.977 | 0.524 | 0.812 | 0.962 | 0.949 | 0.988 | 0.964 | 0.569 |

Furthermore, water was not detected in the hyperspectral imagery when the classification was performed using the previously applied SAM technique with the extracted hyperspectral endmember values. The classification using endmembers achieved an accuracy of 53.41% (Figure 6). Similarly, in the case of multispectral imagery, the classification accuracy using endmembers was 41.61%. These results highlight the limitations of im-

age classification using endmembers in both hyperspectral and multispectral datasets, particularly when it comes to accurately identifying water areas.

**Table 7.** Correlation analysis results between endmember and pixel sampling data using multispectral images.

|  | Class 1 | Class 2 | Class 3 | Class 4 | Class 5 | Class 6 |
|---|---|---|---|---|---|---|
| Reed | 0.339 | 0.95 | 0.942 | 0.899 | 0.614 | 0.878 |
| Water | −0.101 | −0.808 | −0.745 | −0.882 | −0.228 | −0.505 |
| Road | 0.551 | 0.895 | 0.823 | 0.809 | 0.778 | 0.987 |
| Sand | 0.351 | 0.951 | 0.943 | 0.896 | 0.615 | 0.876 |
| Pine | 0.358 | 0.982 | 0.927 | 0.983 | 0.505 | 0.792 |
| Grass | 0.348 | 0.98 | 0.937 | 0.974 | 0.511 | 0.796 |
| Mud | 0.131 | −0.603 | −0.674 | −0.7 | −0.006 | −0.224 |
| Maple | 0.299 | 0.967 | 0.938 | 0.937 | 0.542 | 0.831 |
| Pergola | 0.451 | 0.893 | 0.858 | 0.916 | 0.473 | 0.725 |
| Cherry | 0.368 | 0.988 | 0.932 | 0.977 | 0.524 | 0.812 |

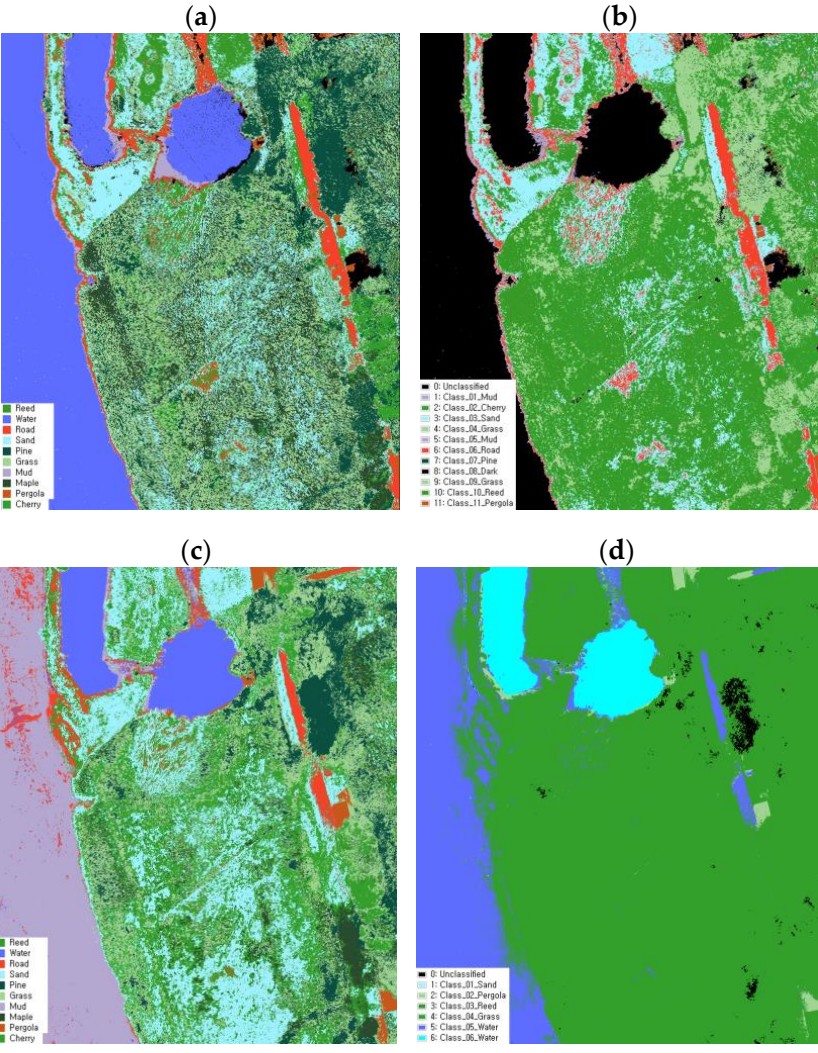

**Figure 6.** SAM mapping using multispectral and hyperspectral images: (**a**) pixel sampling classification of hyperspectral images; (**b**) end member classification of hyperspectral images; (**c**) pixel sampling classification of multiplexed images; and (**d**) end member classification of multiplexed images.

## 4. Discussion

*4.1. Comparison and Evaluation of Hyperspectral and Multispectral Data*

The multispectral and hyperspectral data obtained using UAVs have been used in many studies, but data analysis using training data construction shows the need for improvement. When constructing land cover maps using multispectral and hyperspectral light, it is difficult to simultaneously mount two UAV sensors into one body, which shows that there is a limit to the supply and demand of drone power and available weight. Therefore, time resolution problems can occur when acquiring images, and dissimilarities are identified between multispectral images and hyperspectral images, even though they are taken on the same day [73]. In this paper, hyperspectral images were photographed from 12:30 to 14:00, and multispectral images were photographed from 15:30 to 16:30. Both successfully completed on the same day, but a time error was confirmed when acquiring data using Rededge-MX. As a result, the difference in the left water system part of the multispectral image was confirmed, and it is judged to be due to the amount of light and angular height of the sun. Due to the aforementioned problems, the multispectral classification was organized into stones and mud, and in this study, it was substantiated that the road grade was composed of concrete and the rock type was similar to the road. When verifying data through image classification, it was confirmed that five of the verification target sites corresponding to No. 1 to No. 5 were less accurate. Each area is a verification destination with complex components as compared to other destinations, such as roads, autumn leaves, cherries, and grass, which indicates that categorization using hyperspectral light is more accurate than using multispectral light with fewer classes. In particular, the accuracy of the road was high, and in the case of autumn leaves, it is validated that the accuracy is high where the characteristics due to red light are clear. However, an overall high precision was recognized, but complex components, especially tree classes where overlap is identified, are less accurate in grouping, which results in other division algorithms being acknowledged as having the same problem [74]. Furthermore, if the reflectance of light from the sun, such as on sand, is high, roads are pinpointed with a similar spectrum.

*4.2. Evaluation of Sampling Techniques*

In this paper, we conducted a comparison of data sampling methods when classifying images. Although endmember extraction was used as an effective underlying data sampling technique in hyperspectral studies, it is difficult to obtain pure pixels of images from classifications within a similar spectrum, confirming that only some characteristic arrangements are possible. Similar data, such as for plants, suggest that there is a limit to detecting a complex target spectrum [75]. The endmember technique is considered particularly difficult to apply to multispectral images. This is evidenced by the fact that there are forty-eight maximum selection classes for hyperspectral light but only six for multispectral light, which varies from the absolute number. When comparing multispectral genres, problems are identified not only in the limitations of the six classes, but also in the degree of separation between the classes. Class 6 is highly associated with roads, whereas classes 2, 3, and 4 are determined to have a too low class separation from plants such as cherries, pine trees, reeds, and autumn leaves to pergolas and roads. This finding indicates that it is not effective for the correct class separation. Although we selected 11 classes with the same hyperspectral intensity and high association for spectral features, they were mainly distributed in classes 2, 3, 9, and 10, especially for statistical correlations above 0.9. The components corresponding to the area of many sheaths were mainly composed of grass, sand, reeds, and pine trees. Therefore, we discovered that sampling techniques using endmember extraction were not effective in classifying complex land coverings. The pixel sampling method was more effective for classification than the endmember technique, and pixel sampling had the advantage of reducing the phenomenon of "salt and pepper noise" [76]. Therefore, it is confirmed that pixel sampling techniques enable effective noise reduction and spectral library construction for class types. In this paper, when the land

cover map was produced using the final member, the OA results were confirmed to be 53.41% for hyperspectral images and 41.61% for multispectral images.

### 4.3. Assessment of Classification Techniques

We evaluated the classification results and methodology by building a quantitative spectral library of images using pixel sampling. Among them, we illustrated that the SAM classification outperformed the other classification methods in terms of classification image analysis with an accuracy of 91.906%. Comparing the ANN, SVM, and SAM methods, SAM indicated a difference of up to 16.08% for multispectral images and up to 7.19% for hyperspectral images compared to ANN and SVM. With the development of many classification techniques, the evaluation was conducted using each classification method in the study of land cover evaluation. The authors of [76] reported an OA of 87.75% for RF, 83.31% for CNN, and 80.29% for SVM due to mapping using the Random Forest (RF), CNN, and SVM models. In [77], the classification of urban areas was evaluated using ANN, SVM, ML, and SAM, and the overall accuracies obtained were 92.33%, 85.86%, 83.41%, and 46.55%, with Kappa coefficients of 0.91, 0.83, 0.80, and 0.38, respectively. The ANN method was reported to be effective. The authors of [78] conducted plant species mapping in alpine meadows and confirmed the accuracy of RF as 94.27%, SVM as 89.94%, ANN as 93.51%, and SAM as 78.62%, demonstrating the effectiveness of the RF method. However, these methods need to be simplified because they are complicated to use due to difficulties in approaching and training a large quantity of data. Because the classification performance of the SAM was determined in many studies and demonstrated low accuracy in most papers, the resulting data may depend on the number of samples and data characteristics, but the classification accuracy primarily hinges on the maximum angle threshold [79].

In this paper, we established an effective classification method by examining the classification accuracy of the SAM, SVM, and ANN using both multispectral images and hyperspectral images. However, the lack of field data and the difficulty of validating the entire region limited the building of spectral libraries. To address these limitations, additional validation data and testing are required for the quantification of different layers. In addition, quantitative data, such as ground-based spectrometer measurements, may contribute to more effective data acquisition. The analysis of hyperspectral images revealed the importance of sampling because the number of bands and spectral resolution increased as compared to multispectral images, resulting in the detection of many endmembers. However, the difference between pixel sampling and end member extraction was realized to be significant, and multispectral images showed more uniformity when utilizing more hyperspectral images. As a result, the hyperspectral images have distinct spectral properties for in-image classification, highlighting their effectiveness. Even without ground data and reference spectra, the classification accuracy of spectral images exceeded at least 85% regardless of the technique adopted. Our results suggest that high-resolution hyperspectral images can be used to provide an effective approach on a larger spatial and temporal scale.

### 5. Conclusions

This study sought to achieve effective wetland mapping by applying various classification methods based on UAVs. Mapping wetlands accurately using multispectral and hyperspectral images poses significant challenges. However, through the normalization of a wetland map creation, our study validated the effectiveness of establishing a spectrum library through pixel sampling and comparing it with endmember extraction. The accuracy of the spectrum library was further verified by comparing it with actual field verification data using SAM, SVM, and ANN classification. The results demonstrate the potential for effective data mapping in wetlands. Using the SAM methodology, hyperspectral images achieved an accuracy of 91.91%, whereas multispectral images achieved an 80.36% accuracy. Compared to other methods, the SAM method showed the highest effect with a difference of about 10% in multispectral data and about 7.15% in hyperspectral data. In addition, the effectiveness of the pixel-based sampling method was confirmed, and an accuracy

difference of up to 38.62% was confirmed compared to the final member and sampling method. Future work should focus on evaluating the classification accuracy using multiple target sites, incorporating ground spectrometers, and applying pixel sampling techniques with a larger number of data samples. This study contributes to the literature by proposing an effective data sampling and classification technique evaluation not only for wetland mapping but also for other spectral image applications. Moreover, our findings highlight the importance of developing effective classification methods for the accurate mapping and monitoring of various environments.

**Author Contributions:** Conceptualization, T.Y.L.; methodology, T.Y.L.; software, T.Y.L.; validation, T.Y.L. and W.S.; formal analysis, T.Y.L.; investigation, T.Y.L. and J.K.; writing—original draft, T.Y.L.; writing—review and editing, T.Y.L.; supervision, W.K. and W.S.; project administration, T.Y.L. and W.S. All authors have read and agreed to the published version of the manuscript.

**Funding:** This work was supported by the Korea Environment Industry & Technology Institute (KEITI) through the Exotic Invasive Species Management Program, funded by the Korea Ministry of Environment(MOE) (2021002280001).

**Data Availability Statement:** The data presented in this study are available on request from the corresponding author. The data are not publicly available due to personal information protection and research ethics.

**Conflicts of Interest:** The authors declare no conflict of interest.

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
