# Peer review of "A Study on Wetland Cover Map Formulation and Evaluation Using Unmanned Aerial Vehicle High-Resolution Images"

_drones, doi:10.3390/drones7080536_

Round 1
Reviewer 1 Report
Drone acquired multispectral and hyperspectral imagery are increasingly used in environmental study, particularly land surface mapping. This study compared the accuracy of three machine learning algorithms (SAM, SVM and ANN) using multispectral and hyperspectral data. The authors also compared the performance of two sampling methods (endmember and pixel sampling). The authors found that 1) pixel sampling is superior to endmember; and 2) SAM is better than the other two approaches. This is an interesting study and great potential for wide application. There are three major issues should be addressed:
1) The structure of the manuscript: there should be a section of Discussion or section 3 should be Results and Discussions. Actually, Line 332 -382 can be converted Discussion if adding comparison with other studies in the field.
2) Random forest is often used (or even dominant) in land cover classification – this approach should be included in any comparison study.
3) Performance metrics: there are many metrics other than overall accuracy and Kappa, and some researchers find Kappa is not suitable for assessing classification (see the literatures below).
In addition, the draft needs a thorough English editing, some examples:
Ln 11, coating type should be land cover type
Ln 27, remove “herein”
Ln 29: remove “collectively”
Ln 47: Remote-based surveys
Ln 56: “diverse materials” could be “diverse surface textures”
Ln 272, 274: remove “refer to”
Ln 313: reference to figures should be consistent, either Fig. X or Figure X.
References:
Pontius Jr, R. G., & Millones, M. (2011). Death to Kappa: Birth of quantity disagreement and allocation disagreement for accuracy assessment. International Journal of Remote Sensing, 32(15), 4407–4429. https://doi.org/10.1080/01431161.2011.552923
Foody, G. M. (2020). Explaining the unsuitability of the kappa coefficient in the assessment and comparison of the accuracy of thematic maps obtained by image classification. Remote Sensing of Environment, 11.
Pontius Jr, R. G. (2019). Component intensities to relate difference by category with difference overall. International Journal of Applied Earth Observation and Geoinformation, 77, 94–99. https://doi.org/10.1016/j.jag.2018.07.024
See above
Author Response
Drone acquired multispectral and hyperspectral imagery are increasingly used in environmental study, particularly land surface mapping. This study compared the accuracy of three machine learning algorithms (SAM, SVM and ANN) using multispectral and hyperspectral data. The authors also compared the performance of two sampling methods (endmember and pixel sampling). The authors found that 1) pixel sampling is superior to endmember; and 2) SAM is better than the other two approaches. This is an interesting study and great potential for wide application. There are three major issues should be addressed:
- Thank you for your sincere feedback on the manuscript.
- The structure of the manuscript: there should be a section of Discussion or section 3 should be Results and Discussions. Actually, Line 332 -382 can be converted Discussion if adding comparison with other studies in the field.
- Thank you for your careful consideration. I wrote a part about the discussion and revised it by adding comparisons with other papers. For the land cover classification, we revised 4. Discussion part by referring to other methods as well as random forest. The part you pointed out has been modified to random forest review of paper.
2) Performance metrics: there are many metrics other than overall accuracy and Kappa, and some researchers find Kappa is not suitable for assessing classification (see the literatures below).
- For overall accuracy, the accuracy using the confusion matrix was confirmed, and in addition to Kappa, F1 score were added.
3) In addition.the draft needs a thorough English editing, some examples:
Ln 11, coating type should be land cover type
- Thank you for your detailed feedback. This part has been modified as follows.
- However, the classification of complex
Ln 29: remove “collectively”
- Thank you for your detailed feedback. This part has been modified as follows.
- Our findings provide insights into the accuracy and classification outcomes of different models based on the sampling method employed in spectral imagery.
Ln 47: Remote-based surveys
- Thank you for your detailed feedback. This part has been modified as follows.
- Remote-based surveys allow for non-invasive approaches and quantitative data acquisition.
Ln 56: “diverse materials” could be “diverse surface textures”
- Thank you for your detailed feedback. This part has been modified as follows.
- Mapping research utilizing high-resolution spectral data, such as multispectral and hy-perspectral, proves valuable in monitoring wetlands composed of diverse surface textures.
Ln 272, 274: remove “refer to”
- Thank you for your detailed feedback.
- Added references to additional analysis methods for accuracy verification.
Ln 313: reference to figures should be consistent, either Fig. X or Figure X.
- Thank you for your detailed feedback.
- We have explained and referred to Table 5.
References:
Pontius Jr, R. G., & Millones, M. (2011). Death to Kappa: Birth of quantity disagreement and allocation disagreement for accuracy assessment. International Journal of Remote Sensing, 32(15), 4407–4429. https://doi.org/10.1080/01431161.2011.552923
Foody, G. M. (2020). Explaining the unsuitability of the kappa coefficient in the assessment and comparison of the accuracy of thematic maps obtained by image classification. Remote Sensing of Environment, 11.
Pontius Jr, R. G. (2019). Component intensities to relate difference by category with difference overall. International Journal of Applied Earth Observation and Geoinformation, 77, 94–99. https://doi.org/10.1016/j.jag.2018.07.024

Reviewer 2 Report
Introduction
The Introduction needed to be restructured.
Methods
Line 103- Is that 126 degree and 84.3 minutes? But 60 minutes equals to 1 degree.
Line 109-110- Is it because this specific area contains various land cover?
Results
Line 274-277- Authors states that SAM is the best classifier when using multispectral data. I am wondering that why there were some roads showed in the west part of the classified result of SAM-Multispectral, which is supposed to be water or mud. The accuracy of SVM was 71.43%, and this was very slightly less than SAM. However, in the map of SVM-Multispectral, the area of Pine or Maple was notable, and also more Pergola, compared to SAM. These two maps showed large difference visually, but the accuracies were similar. It seems that authors have evaluated the effectiveness only based on the numerical accuracy, but did not care about the area or distribution of the species or land cover. Further, it is difficult to distinguish Reed and Cherry, Pine and Maple in Figure 5.
Line 296-297- Which coefficient was significant? It should be marked in the Tables.
Line 307- What is ‘superspectral’?
Line 332- Is this the start of the Discussion? In this section, it should not repeat in detail what you have done, but interpret and discuss your findings by comparing with previous researches.
Author Response
Introduction
- The Introduction needed to be restructured.
- Thank you for your sincere feedback on the manuscript.
- I modified the 1. introduction part of the paper.
Methods
- Line 103- Is that 126 degree and 84.3 minutes? But 60 minutes equals to 1 degree.
- Thank you for your detailed feedback. This part has been modified as follows.
- This study was conducted at Ansan Reed Wetland, located in Ansan-si, Korea. The wetland is situated at 37°16'21.32N, 126°50'21.64E, with a total area of 1,037,500 m2
- Line 109-110- Is it because this specific area contains various land cover?
- In the paper, we conducted a UAV flight with sensors installed on the Ansan reed wetland. Among them, many problems occurred during the shooting, such as bad pixels and wind problems caused by light scattering, and during the field survey, we found a place with high image quality in the video and selected it as a research target.
- The structure of the manuscript: there should be a section of Discussion or section 3 should be Results and Discussions. Actually, Line 332 -382 can be converted Discussion if adding comparison with other studies in the field.
- The content was revised by writing the discussion in Part 4.
Results
- Line 274-277- Authors states that SAM is the best classifier when using multispectral data. I am wondering that why there were some roads showed in the west part of the classified result of SAM-Multispectral, which is supposed to be water or mud.
- This part was written by supplementing the parts discussed in Part 4. It is judged by the difference in time resolution between multi-spectral images and hyperspectral images and the change in the water body reflectance according to the solar altitude angle.
- The accuracy of SVM was 71.43%, and this was very slightly less than SAM. However, in the map of SVM-Multispectral, the area of Pine or Maple was notable, and also more Pergola, compared to SAM. These two maps showed large difference visually, but the accuracies were similar. It seems that authors have evaluated the effectiveness only based on the numerical accuracy, but did not care about the area or distribution of the species or land cover.
- Thank you for your detailed feedback.
- Created zone table for classified classes. This has been modified to determine the class ratio according to the analysis model.
- Further, it is difficult to distinguish Reed and Cherry, Pine and Maple in Figure 5.
- The color of the tree was given by referring to the classification color value (https://egis.me.go.kr/intro/land.do) ) provided in Korea. You have created zone tables for classified classes for better visibility of tree classification.
- Line 296-297- Which coefficient was significant? It should be marked in the Tables.
- Significant coefficients are shown in Table 3 and are added to the results.
- Line 307- What is ‘superspectral’?
- Thank you for your detailed feedback. This part has been modified as follows.
- The content has been modified to "Hyperspectral".
- Line 332- Is this the start of the Discussion? In this section, it should not repeat in detail what you have done, but interpret and discuss your findings by comparing with previous researches.
- Thank you for your careful consideration. I wrote a part 4. discussion and revised it by adding comparisons with other papers.

Reviewer 3 Report
This review provides feedback on the manuscript titled " A study on wetland cover map formulation and evaluation using UAV high-resolution images." The paper explores an interesting topic that holds potential value for future readers. After a thorough examination of the manuscript, I have the following observations:
1.Here is some of the similar work which has been done.
DOI: 10.5194/isprsarchives-XLI-B1-781-2016,
Quantitative assessment of urban wetland dynamics using high spatial resolution satellite imagery between 2000 and 2013 | Scientific Reports (nature.com)
High-resolution mapping based on an Unmanned Aerial Vehicle (UAV) to capture paleoseismic offsets along the Altyn-Tagh fault, China | Scientific Reports (nature.com)
These are the work which already exist. Please consult this paper and see how your work add value to these.
2.Novelty of this work need to be highlighted in order to make this work more useful for future readers.
3.The overall paper is not written properly, different section does not adequately cover the required details. Part of introduction added in method. It required to be improved further by adding more of literature.
4.There are optical and hyperspectral data is available in public domain, author does not consider the comparing them with UAV data. This will help the future reader to understand the benefit of using UAV data. In the following link you can see the Optical free data https://sentinel.esa.int/web/sentinel/sentinel-data-access and for hyperspectral free data https://www.eoportal.org/satellite-missions/prisma-hyperspectral#prisma-hyperspectral-precursor-and-application-mission
5.There is mention about the geometric and radiometric correction but it does not sufficient details. Which DEM were used for the correction? There is no flowchart added showing various data processing steps.
6.The overall it looks like simple project which has utilised the standard software such as Pix4D, ENVI, ARCGIS and Trimble software. this work does not highlight the what is contribute to the scientific community.
I believe that considering these comments in a constructive manner will contribute to enhancing the quality of this work for the benefit of a wider audience. Thank you.
Author Response
This review provides feedback on the manuscript titled " A study on wetland cover map formulation and evaluation using UAV high-resolution images." The paper explores an interesting topic that holds potential value for future readers. After a thorough examination of the manuscript, I have the following observations:
- Thank you for your sincere feedback on the manuscript.
1) Here is some of the similar work which has been done.
DOI: 10.5194/isprsarchives-XLI-B1-781-2016,
Quantitative assessment of urban wetland dynamics using high spatial resolution satellite imagery between 2000 and 2013 | Scientific Reports (nature.com)
High-resolution mapping based on an Unmanned Aerial Vehicle (UAV) to capture paleoseismic offsets along the Altyn-Tagh fault, China | Scientific Reports (nature.com)
These are the work which already exist. Please consult this paper and see how your work add value to these.
- Thank you for your sincere feedback on the manuscript.
- I will add part 4. Discussion contents by referring to the reference.
2) Novelty of this work need to be highlighted in order to make this work more useful for future readers.
- To emphasize novelty, I will add parts 1, 4, and 5.
3) The overall paper is not written properly, different section does not adequately cover the required details. Part of introduction added in method. It required to be improved further by adding more of literature.
- Thank you for your sincere feedback on the manuscript.
- Added references and created part 4 to improve.
4) There are optical and hyperspectral data is available in public domain, author does not consider the comparing them with UAV data. This will help the future reader to understand the benefit of using UAV data. In the following link you can see the Optical free data https://sentinel.esa.int/web/sentinel/sentinel-data-access and for hyperspectral free data https://www.eoportal.org/satellite-missions/prisma-hyperspectral#prisma-hyperspectral-precursor-and-application-mission
- Thank you for your sincere feedback on the manuscript.
- Complex regional classifications, such as wetlands, have been put on hold because the quality of public data is not yet high, and existing studies have shown clear differences between satellite data and UAV data. This is a problem with pixel resolution and has been demonstrated in existing satellite classification studies, and satellite analysis is more likely to have problems with verification and is highly accurate in most studies. Utilizing UAVs has great advantages and requires high-resolution-based regional features.
- Part 4. Add these to the discussion section to highlight the novelty of the research.
5) As the quality of public data increases, it is expected to be as high as existing UAVs someday, so in this study, the sampling method can be used as basic data for classification research, and various data combinations are required. There is mention about the geometric and radiometric correction but it does not sufficient details. Which DEM were used for the correction? There is no flowchart added showing various data processing steps.
- Details of the image data processing process of the UAV have been added.
- The DEM used in the Preprocessing is a DEM using a Velodyne rider that is optionally attached to the hyperspectral light (https://www.headwallphotonics.com/products/vnir-400-1000nm) . This DEM can be taken simultaneously with hyperspectral photography, so it can act effectively during geometric correction.
6) The overall it looks like simple project which has utilised the standard software such as Pix4D, ENVI, ARCGIS and Trimble software. this work does not highlight the what is contribute to the scientific community.
- Leveraging standard software seeks to extend the accessibility of the methodology. In addition, in this paper, by comparing sampling methods and classification algorithms, these methods make effective access to the public.
- Part 4. In the discussion section, I will add an additional story about the utilization of commercialized software.
7) I believe that considering these comments in a constructive manner will contribute to enhancing the quality of this work for the benefit of a wider audience. Thank you.
- Thank you for your sincere feedback on the manuscript.

Round 2
Reviewer 1 Report
The reversion has addressed my comments sufficiently. I am happy to sign off my review although there are some English grammar and expression errors need to be corrected, such as:
Ln 397, In the study of the pixel sampling method authenticated a decrease in the "salt and pepper" phenomenon - confusing;
Ln 416. The author of was evaluated using SVM, RF, ANN, and SAM when classifying plant species ... - The author of what?
See above
Author Response
The reversion has addressed my comments sufficiently. I am happy to sign off my review although there are some English grammar and expression errors need to be corrected, such as:
Re: Thank you for your sincere feedback on the manuscript.
1) Ln 397, In the study of the pixel sampling method authenticated a decrease in the "salt and pepper" phenomenon - confusing;
Re: The part you pointed out has been modified to " Pixel sampling method is more effective for classification than Endmember, and pixel sampling has the advantage of reducing the phenomenon of "salt and pepper noise" [76].
Therefore, it is confirmed that pixel sampling techniques enable effective noise reduction and spectral library construction for class types. In this paper, when the land cover map was produced using the final member, OA confirmed the results of 53.41% hyperspectral images and 41.61% multispectral images.
Mapping Wetland Plant Communities Using Unmanned Aerial Vehicle Hyperspectral Imagery by Comparing Object/Pixel-Based Classifications Combining Multiple Machine-Learning Algorithms
2) Ln 416. The author of was evaluated using SVM, RF, ANN, and SAM when
classifying plant species ... - The author of what?
Re: Thank you for your sincere feedback on the manuscript.
The part you pointed out has been modified to “The authors of [78] conducted plant species mapping in alpine meadows and confirmed the accuracy of RF 94.27%, SVM 89.94%, ANN 93.51%, and SAM 78.62%, demonstrating the effectiveness of the method used in RF.”

Reviewer 2 Report
The authors have addressed all my concerns. I have no further comments. Please check grammar mistakes carefully.
Author Response
The authors have addressed all my concerns. I have no further comments. Please check grammar mistakes carefully.
Re: Thank you for your detailed feedback.
Reviewer 3 Report
Dear Author,
Thanks for the reply, but still there is work need to be done.
best wishes,
Author Response
Dear Author,
Thanks for the reply, but still there is work need to be done.
best wishes,
Re: Thank you for your careful review. The text has been amended so that it can be explained correctly.